# The mTOR Inhibitor Rapamycin Counteracts Follicle Activation Induced by Ovarian Cryopreservation in Murine Transplantation Models

**DOI:** 10.3390/medicina59081474

**Published:** 2023-08-16

**Authors:** Jules Bindels, Marlyne Squatrito, Laëtitia Bernet, Michelle Nisolle, Laurie Henry, Carine Munaut

**Affiliations:** 1Laboratory of Biology of Tumor and Development, GIGA-Cancer, Université de Liège, 4000 Liège, Belgium; jules.bindels@uliege.be (J.B.); marlyne.squatrito@uliege.be (M.S.); laetitia.bernet@uliege.be (L.B.); 2Department of Obstetrics and Gynecology, Hôpital de la Citadelle, Université de Liège, 4000 Liège, Belgium; michelle.nisolle@chuliege.be (M.N.); laurie.henry@citadelle.be (L.H.)

**Keywords:** ovarian cryopreservation, follicle activation, fertility preservation, heterotopic transplantation, animal models

## Abstract

*Background and Objectives*: Ovarian tissue cryopreservation followed by autotransplantation (OTCTP) is currently the only fertility preservation option for prepubertal patients. Once in remission, the autotransplantation of frozen/thawed tissue is performed when patients want to conceive. A major issue of the procedure is follicular loss directly after grafting mainly due to follicle activation. To improve follicular survival during the OTCTP procedure, we inhibited the mTOR pathway involved in follicle activation using rapamycin, an mTOR inhibitor. Next, we compared two different in vivo models of transplantation: the recently described non-invasive heterotopic transplantation model between the skin layers of the ears, and the more conventional and invasive transplantation under the kidney capsule. *Materials and Methods*: To study the effects of adding rapamycin during cryopreservation, 4-week-old C57BL/6 mouse ovaries, either fresh, slow-frozen, or slow-frozen with rapamycin, were autotransplanted under the kidney capsule of mice and recovered three weeks later for immunohistochemical (IHC) analysis. To compare the ear with the kidney capsule transplantation model, fresh 4-week-old C57BL/6 mouse ovaries were autotransplanted to either site, followed by an injection of either LY294002, a PI3K inhibitor, vehicle control, or neither, and these were recovered three weeks later for IHC analysis. *Results*: Rapamycin counteracts cryopreservation-induced follicle proliferation, as well as AKT and mTOR pathway activation, in ovaries autotransplanted for three weeks under the kidney capsule of mice. Analyses of follicle proliferation, mTOR activation, and the effects of LY294002 treatment were similar in transplanted ovaries using either the ear or kidney capsule transplantation model. *Conclusions*: By adding rapamycin during the OTCTP procedure, we were able to transiently maintain primordial follicles in a quiescent state. This is a promising method for improving the longevity of the ovarian graft. Furthermore, both the ear and kidney capsule transplantation models were suitable for investigating follicle activation and proliferation and pharmacological strategies.

## 1. Introduction

In recent decades, the chances of surviving cancer have substantially increased thanks to the improvement of cancer therapies. As a result of this, considering the quality of life after remission has become increasingly important. One of the important issues among young cancer survivors hoping to one day become parents is the ability to have biological children. Unfortunately, certain chemo- and radiotherapies (such as alkylating agents) can increase the risk of ovarian failure and infertility due to gonadotoxic effects, removing the possibility of natural pregnancies [1,2].

One of the available fertility preservation options is the cryopreservation of cortical ovarian tissue followed by autotransplantation (OTCTP). At present, for prepubertal girls and young women who require urgent oncological treatment, OTCTP is the only effective fertility preservation technique [3]. Currently, more than 130 live births have taken place after the use of OTCTP, with a live birth rate between 25 and 42% and mean graft longevity of 27 months [4,5,6,7,8]. A major advantage of OTCTP is the possibility to perform the preservation at any moment of the menstrual cycle without delaying the oncological treatment, as compared to, e.g., mature oocyte cryopreservation [9]. Furthermore, it can restore both the endocrine function of the gonads and natural fertility, and it is not limited to a single pregnancy but allows for multiple births [10,11]. However, a major issue of this technique is the loss of follicles directly after grafting. Indeed, multiple studies have shown a decrease in primordial follicles in transplanted ovaries compared to non-transplanted ovaries [12,13,14]. The depletion of the follicle reserve, reducing the longevity of transplants, is possibly due to transient ischemia, apoptosis, and/or massive follicle recruitment, which is known as follicular “burn-out” [15]. Previous approaches to enhance graft longevity, e.g., enhancing revascularization or limiting apoptosis [16,17,18,19,20], have led to some but still insubstantial improvements in graft survival [21,22]. However, recent studies tested an ovarian tissue transplantation mice model that used adipose-tissue-derived stem cells, and the results showed substantially enhanced graft vascularization and the increased quiescence of primordial follicles, leading to a larger follicle pool both after short- and long-term grafting [23,24,25,26]. In order to increase graft longevity, which eventually leads to enhanced pregnancy rates after the use of OTCTP, additional studies are still required to find new approaches that limit follicle “burn-out”.

Previous research showed the involvement of multiple molecular pathways, such as phosphatidylinositol-3-kinase (PI3K)/phosphatase and tensin homolog (PTEN)/AKT and the mammalian target of rapamycin (mTOR) pathways, in follicle activation; the manipulation of these pathways in cultured ovaries resulted in a significant change in primordial follicle levels compared to control ovaries [27,28]. Under normal physiological conditions, the primordial follicle reserve is maintained via the complex balance between activation signals provided by, but not limited to, the PI3K/PTEN/AKT and mTOR pathways and inhibition via the anti-Müllerian hormone (AMH) secreted by growing follicles. However, due to ovarian cryopreservation followed by autotransplantation, this balance is dysregulated; follicle inhibition signals are diminished as many growing follicles do not survive freezing and thawing. This leads to an imbalance towards follicle activation, eventually resulting in massive follicle recruitment and thus a decrease in the follicle reserve [15,29,30].

Currently, different inhibitors of the follicle activation pathways have been tested on ovaries in various experimental settings. The PI3K inhibitor LY294002 (LY) was investigated both in vitro and in vivo, and the results showed that it could inhibit ovarian carcinoma cell growth [31]. In vivo studies with the natural follicle inhibitor, AMH, have shown controversial results on follicle preservation when injected into mice before or after OTCTP [32,33]. However, a recent study showed that mice administrated with AMH after ovarian transplantation had a higher primordial follicle reserve compared to mice injected with the control [34]. Other studies focused on a specific mTOR inhibitor, rapamycin, which is already used in clinics to, e.g., prevent rejection after organ transplantation [35,36]. Rapamycin has been shown to decrease the ratio of growing follicles to primordial follicles in cultured cisplatin-treated fresh rat ovaries [35]. Furthermore, it was observed that the injection of rapamycin in rats could preserve the primordial follicle pool [37], and SCID mice transplanted with vitrified human ovarian tissue followed by rapamycin injection showed a higher percentage of remaining primordial follicles compared to control mice [38]. So far, these promising results with rapamycin were obtained using fresh or vitrified ovaries. To be used in a clinic, a study focusing on slow-frozen (SF) ovaries is required, which is currently the gold-standard ovarian tissue cryopreservation method [39].

Our previous research showed that the slow-freezing of ovaries induced follicle activation via the PI3K/PTEN/AKT and mTOR pathways, which could be counteracted by adding LY294002 or rapamycin to the freezing medium. Furthermore, the culture-induced activation of follicle activation pathways could be counteracted by cryopreservation with rapamycin and culture with LY294002 [40]. As these in vitro results with rapamycin were very promising, our first aim was to confirm in vivo that rapamycin could counteract cryopreservation- and transplantation-induced follicle activation using the widely used experimental ovarian transplantation model, namely, transplanting ovaries under the kidney capsule of mice [41,42].

Many different ovarian transplantation sites have been investigated to test pharmacological strategies, with every site having its advantages [13,41,42,43,44,45,46,47,48]. Our team previously evaluated an alternative heterotopic transplantation site for ovarian grafts between the skin layer and cartilage of the ears, adapted from the ear sponge assay that was set up to study (lymph) angiogenesis [49,50]. This site was chosen as the transplantation procedure is less invasive compared to more conventional models. Additionally, the ears are highly vascularized, being of great importance as major follicle “burn out” has been reported due to ischemia [12,13,14]. Our second aim was to compare this new model to the more conventional kidney capsule transplantation site.

## 2. Materials and Methods

### 2.1. Experimental Design

The experimental designs can be found in Figure 1. For the first aim, female C57BL/6 mice (*n* = 27, 4 weeks old) were obtained from Charles River Laboratories (France) and maintained at the accredited Mouse Facility and Transgenics GIGA platform of the University of Liège (Belgium). The mice were kept at ±21 °C in a 12 h light/dark cycle with a maximum of 5 mice per cage.

Ovaries collected from these mice, either fresh, SF with control medium (SFct), or SF with rapamycin (SFra), were autotransplanted under the kidney capsule (*n* = 18 ovaries per group with 2 grafts per mouse).

For the second aim, fresh ovaries from C57BL/6 mice (*n* = 10, 4 weeks old) were autotransplanted either under the kidney capsule or between the skin layers of the ears (*n* = 7–10 ovaries per group with 2 grafts per mouse). Furthermore, as a follow-up to test the treatment administration after transplantation, autotransplantation of fresh C57BL/6 mice ovaries was performed (*n* = 20, 4 weeks old) to either transplantation site, followed by injection with LY294002 or vehicle control locally for ovaries transplanted into the ear (injected in both ears), or systemically (intraperitoneally; IP) for ovaries transplanted under the kidney capsule (*n* = 8–10 ovaries per group with 2 grafts per mouse; injections given every other day with a total of 4 injections; 16.67 mg/kg LY294002 per injection). The weight and behavior of the mice were monitored for a total of 3 weeks to assess the side effects of LY294002 treatment. Behavior scoring was adapted from Herrmann et al. and performed as described in Table 1 [51].

Grafted ovaries from all experiments were collected 3 weeks after transplantation and fixed in 4% formaldehyde overnight, after which ovaries were put in 70% ethanol. Fixed ovaries were embedded in paraffin and cut into 5 µm sections using a microtome and mounted on slides for histological assessment.

The Animal Ethics Committee of the University of Liège approved this study (#1934) and all experiments were performed in accordance with relevant guidelines and regulations.

### 2.2. Oophorectomy and Preparation of Ovaries

For all experiments, bilateral oophorectomy was performed on the mice under gas anesthesia (Isoflurane, Dechra, Northwich, UK), and the ovaries were placed in Leibovitz L-15 medium (Lonza, Verviers, Belgium) supplemented with 10% Fetal Bovine Serum (FBS; Thermo Fisher Scientific, Gibco, Waltham, MA, USA), a transport solution. The oviduct and fat tissue taken during oophorectomy were removed from the ovaries under a binocular microscope using a scalpel. Fresh ovaries were then directly autotransplanted, and ovaries that were going to be slow-frozen were prepared accordingly.

### 2.3. Ovarian Slow-Freezing and Thawing Method

Designated ovaries were SF and thawed as described before [52]. Briefly, whole ovaries were put in a freezing solution containing Leibovitz L-15 medium supplemented with 10% FBS, 10% dimethylsulfoxide (DMSO; Merck, Darmstadt, Germany), and 0.1 M sucrose, and equilibrated for 30 min at 4 °C. After equilibration, ovaries were put in cryovial tubes (Simport, Montreal, QC, Canada) containing freezing solution and thereafter SF in a programmable freezing machine (CL-8800i System; CryoLogic, Mulgrave, Victoria, Australia) as previously described and stored in liquid nitrogen [53]. For ovaries SF with rapamycin (1 μM, InvivoGen, Toulouse, France), the inhibitor was added to the transport and freezing solutions. Thawing was performed by incubating cryovials for 2 min at room temperature (RT) followed by a 2 min incubation at 37 °C in a water bath. Ovaries were then washed in Leibovitz L-15 medium 3 times for 5 min at 37 °C to remove any remaining cryoprotectants and/or rapamycin.

### 2.4. Autotransplantation Procedures

#### 2.4.1. Transplantation under the Kidney Capsule

The transplantation of ovaries under the kidney capsule was based on an article by Nicholson et al. [54]. Briefly, the kidney was exteriorized under gas anesthesia (isoflurane) and kept hydrated by applying saline solution. Using thin-tip forceps, the kidney capsule was carefully lifted from the kidney parenchyma and a small incision was made using tiny spring scissors. A small pocket for the ovaries was created by manipulating a rounded closed-end glass Pasteur pipette under the capsule. Both whole ovaries were then inserted into the pocket using the glass pipette and gently pushed away from the hole to decrease the risk of the ovaries slipping out of the pocket. Once the grafting was complete, the peritoneum was closed using a double suture, and the skin was closed using surgical wound clips.

#### 2.4.2. Transplantation between Skin Layers of the Ears

A minor incision was made in the basal, external, and central parts of the mouse ear under gas anesthesia (isoflurane), and the external skin layer was detached from the cartilage using thin-tip forceps. The whole fresh ovary was placed through the incision between the skin layer and the cartilage, and the skin was subsequently sutured to close the hole [55].

### 2.5. Histological Assessment

In order to perform follicle identification and quantification, ovarian sections were labeled with LIM-homeobox protein 8 (LHX8; Abcam ab41519, Cambridge, UK) transcription factor and/or DEAD-box helicase 4 (DDX4; Abcam ab13840, Cambridge, UK). To analyze follicle proliferation and activation of the AKT and mTOR pathways, sections were labeled for KI67 (Abcam ab16667, Cambridge, UK), phosphor-AKT (pAKT; Abcam ab81283, Cambridge, UK), and phosphor-RPS6 (pRPS6; Cell Signaling #2211, Danvers, MA, USA), respectively. Apoptosis was revealed by immunostaining cleaved caspase-3 (Cell Signaling #9661, Danvers, MA, USA) and TUNEL staining (Roche 11684795910, Mannheim, Germany; following the manufacturer’s instructions). Vascular endothelial cells were evidenced by immunostaining CD31 (Abcam ab28364, Cambridge, UK). Fibrosis was identified using Van Gieson staining.

Briefly, for immunostainings, ovarian sections were deparaffinized and rehydrated, followed by antigen retrieval using an autoclave (11 min, 126 °C, 1.4 Bar) in either citrate (for LHX8, DDX4, pRPS6, KI67, and cleaved caspase-3) or target buffer (for pAKT and CD31) (Dako, Glostrup, Denmark). After cooling down for 20 min, endogenous peroxidase activity was blocked using 3% hydrogen peroxide for 20 min at RT. Non-specific binding sites were blocked by incubation with Animal-Free Blocking Solution (Cell Signaling, Danvers, MA, USA) for 20 min at RT. Primary antibodies (diluted in REAL antibody diluent (Dako, Glostrup, Denmark)) were incubated for 1 h at RT except for cleaved caspase-3, which was incubated overnight at 4 °C (LHX8 1/100; DDX4 1/600; KI67 1/100; pAKT 1/250; pRPS6 1/400; cleaved caspase-3 1/300; CD31 1/200). Afterward, sections were incubated with the secondary antibody linked with horseradish peroxidase (HRP; ENVISION/HRP ready to use, Dako, Glostrup, Denmark) for 30 min at RT. For visible staining, the revelation was performed with DAB+ (Dako, Glostrup, Denmark) followed by hematoxylin counterstaining, and sections were mounted using Entellan new mounting medium (Sigma-Aldrich, St. Louis, MO, USA). For fluorescent staining, the fluorescein tyramide kit (PerkinElmer, Waltham, MA, USA) was used for 10 min and sections were mounted with DAPI FluoromountG mounting medium (SouthernBiotech, Birmingham, AL, USA). Stained sections were then scanned using either the NanoZoomer 2.0 HT digital slide scanner (Hamamatsu Photonics K.K., Hamamatsu, Japan) or the Olympus SLIDEVIEW VS200 high digital slide scanner (Olympus Corporation, Tokyo, Japan).

### 2.6. Follicle Quantification

Scanned sections labeled for LHX8 were analyzed using the NDP.view2 software (Hamamatsu Photonics K.K., Hamamatsu, Japan). At least four to five 5 µm sections per ovary were analyzed blind, with each section being taken 50 µm further down the ovary compared to the previous section. Follicles were classified into primordial, primary, and secondary or more growing according to morphological mouse follicle classification by manually looking at each section and counting and classifying all follicles accordingly [56]. Total follicle density was defined as the number of follicles per mm^2^ (n/mm^2^) after manually outlining the ovarian surface of each section. Results are expressed both as the number of each follicle type per mm^2^ and the percentage of each type relative to the total number of follicles per section. Each section was analyzed individually, followed by calculating the mean of the results of the analyzed sections per ovary.

To analyze follicle proliferation and activation, ovarian sections were double-stained for DDX4 in combination with KI67, pAKT, or pRPS6. Primordial and primary follicles were manually counted using DDX4 staining and classified as stained or non-stained by manually looking at the follicles and determining whether they are stained or not for the desired protein (NDP view software). Results are expressed as the percentage of stained relative to the total primordial and primary follicles. At least two to three 5 µm sections per ovary were analyzed blinded, with each section being taken 50 µm further down the ovary compared to the previous section. Each section was analyzed individually, followed by calculating the mean of the results of the sections per ovary.

### 2.7. Statistical Analysis

GraphPad Prism 8 (GraphPad, San Diego, CA, USA) was used to perform all statistical analyses. The Kruskal–Wallis test with Dunn’s multiple comparison post hoc test was applied when comparing three or more experimental groups. For comparisons between two experimental groups, the Mann–Whitney test was performed. For both tests, *p* < 0.05 was considered statistically significant.

## 3. Results

### 3.1. In Vivo Effects of Adding Rapamycin to the Freezing Medium in Ovaries Heterotopically Transplanted under the Kidney Capsule of Mice

To investigate in vivo whether the addition of rapamycin to the freezing medium can counteract follicle activation and proliferation induced by cryopreservation and/or transplantation, the heterotopic kidney transplantation mice model was used. Immunohistochemical staining was performed to calculate follicle densities and analyze the activation of follicle activation pathways, as well as apoptosis, angiogenesis, and fibrosis.

#### 3.1.1. The Primordial Follicle Pool Is Decreased by Slow-Freezing with or without Rapamycin Compared to Fresh Ovaries

Ovaries were recovered three weeks after transplantation. The follicular densities were analyzed by manually counting primordial, primary, and secondary or more growing follicles using LHX8 labeling (Figure 2A–C). The percentage of primordial follicles, as well as the total number of primordial follicles per mm^2^, was lower when ovaries were SF in the control medium compared to fresh ovaries. The addition of rapamycin to the freezing medium did not reverse this effect. No significant difference in the primary follicle percentage and total number was found in SFct ovaries compared to fresh ovaries. The addition of rapamycin significantly decreased the total number and the percentage of primary follicles compared to fresh ovaries, with no significant difference between SFct and SFra. Furthermore, slow-freezing, both with or without rapamycin, resulted in a significantly higher percentage of secondary or more growing follicles compared to fresh ovaries. However, no difference was observed in the total number of secondary or more growing follicles between the groups (Figure 2D–E).

#### 3.1.2. Addition of Rapamycin to the Freezing Medium Counteracts Follicle Proliferation and Activation Induced by Slow-Freezing and/or Transplantation In Vivo

To further analyze whether adding rapamycin to the freezing medium modulates the cryopreservation/transplantation-induced proliferation and activation of the AKT and mTOR pathways, ovarian sections were stained for KI67, pAKT, and pRPS6, respectively. The percentage of primordial and primary follicles labeled for KI67 displayed a trend toward being higher in SF ovaries in the control medium compared to fresh ovaries. The addition of rapamycin to the freezing medium showed a lower amount of KI67-labeled follicles compared to fresh and SFct conditions (Figure 3A). In addition, significantly more primordial and primary follicles were stained for pAKT (Figure 3B) and pRPS6 (Figure 3C) in SF ovaries with the control medium compared to fresh ovaries. We found that the addition of rapamycin to the freezing medium was able to counteract this effect. Indeed, similar amounts of pAKT- and pRPS6-stained primordial and primary follicles were found in SF ovaries with rapamycin and fresh ovaries three weeks after transplantation.

#### 3.1.3. No Difference in Apoptosis, Vascular Endothelial Cells, and Fibrosis Was Observed between Fresh, SF, or SF Ovaries, with Rapamycin

No apoptosis was observed three weeks after transplantation in the ovarian sections of all experimental groups by active caspase-3 labeling (Appendix A) and by analyzing DNA strand breaks using a TUNEL assay (Appendix A). Furthermore, no difference in the number of vascular endothelial cells (CD31 analysis) was found between the groups (Appendix A). Additionally, Van Gieson staining showed no differences in fibrosis between the three experimental groups (Appendix A).

### 3.2. Comparison of Two Different Ovarian Tissue Transplantation Models

In order to compare the newly described transplantation site between the skin layers of the ears with the more conventional site under the kidney capsule, fresh mice ovaries were autotransplanted to either site with or without LY294002 injection locally in the ears or IP for ovaries transplanted under the kidney capsule.

Mice were monitored for weight gain and their behavior. Transplanted ovaries were recovered after three weeks, and ovarian sections were stained to investigate follicle activation and proliferation and apoptosis.

#### 3.2.1. Graft Vascularization and Follicle Reserve Are Similar between Transplantation Sites, with No Observation of Apoptosis

No significant differences were observed in primordial and primary follicles with respect to the percentage of total follicles as the total number per mm^2^ between the two transplantation sites. A significantly higher percentage of secondary or more growing follicles was observed in ovaries transplanted between the skin layers of the ears compared to ovaries transplanted under the kidney capsule. However, this effect was not observed when examining total secondary or more growing follicles per mm^2^ (Figure 4). The analysis of active caspase-3 showed no apoptotic activity for both transplantation sites (Figure 5A) and no differences in blood vessel formation (CD31 staining) (Figure 5B).

The percentage of primordial and primary follicles labeled for cell proliferation (KI67) (Figure 6A) and the activation of the mTOR pathway (pRPS6) (Figure 6C) was similar between the transplantation sites. However, significantly fewer primordial and primary follicles were labeled with pAKT in ovaries transplanted between the skin layers of the ears compared to ovaries transplanted under the kidney capsule (Figure 6B).

#### 3.2.2. Behavioral Score of Mice after LY294002 Injection in Both Transplantation Models

In order to analyze whether the injection of LY294002 locally in the ear had similar side effects to the IP injection, weight gain and behavioral scores were monitored up to three weeks after injection (Table 1). No differences were observed in weight gain between the four experimental groups after transplantation (Figure 7A). However, the behavioral score was significantly lower for mice systemically (IP) injected with LY294002 compared to mice injected locally in the ears, as well as compared to mice injected with the control (Figure 7B). No differences were observed between the two types of surgery with respect to both weight gain and behavioral score (Figure 7A–B).

#### 3.2.3. Follicle Density, Activation, and Proliferation Are Similar in Both Transplantation Models

Primordial, primary, and secondary or more growing follicle densities, with respect to both the percentage of each type compared to the total amount per section and the total number of each follicle type per mm^2^, were similar between ovaries after the local injection of LY294002 in the ears and IP injection for ovaries transplanted under the kidney capsule. However, IP injection with LY294002 resulted in a higher percentage and total number per mm^2^ of primordial follicles compared to the control injection in the ears and a higher total number of secondary or more growing follicles (Figure 8).

Furthermore, no apoptotic cells were observed in ovarian sections for all four experimental groups (active caspase-3 and TUNEL staining) (Appendix A).

Additionally, local injection in the ears or IP injection did not affect the percentage of primordial and primary follicles stained for KI67 (Figure 9A), pAKT (Figure 9B), or pRPS6 (Figure 9C). However, the percentage of pAKT-stained primordial and primary follicles was significantly lower when mice were injected with LY294002, either locally in the ear or IP, compared to ovaries transplanted under the kidney capsule with the control injection (Figure 9B).

## 4. Discussion

OTCTP is currently the only fertility preservation option for prepubertal patients and young women in need of urgent treatment for severe malignancies [3]. This technique has already led to more than 130 live births, and this number is expected to grow [7,8]. However, the technique still has several limitations. One of the major issues of OTCTP is follicular loss directly after grafting partly due to massive primordial follicle recruitment [15]. Our team previously showed that the slow-freezing of ovaries induced follicle activation via the PI3K/PTEN/AKT and mTOR pathways. Additionally, this could be counteracted by adding LY294002 or rapamycin to the freezing medium. Furthermore, we found that the best combination of inhibitors to counteract the culture-induced activation of follicle activation pathways was adding rapamycin to the freezing medium and performing the culture with LY294002 [40]. These promising in vitro results encouraged us to test adding rapamycin to the freezing medium in vivo using the widely used ovarian transplantation mice model, transplanting ovaries under the kidney capsule [41,42]. Our results indicated that the slow-freezing and thawing of ovaries before transplantation caused follicle proliferation to be higher and increased follicle activation via the PI3K/PTEN/AKT and mTOR pathways compared to the transplantation of fresh ovaries three weeks after transplantation. Furthermore, we showed that the addition of rapamycin to the freezing medium was able to significantly counteract follicle proliferation and activation, thus keeping primordial follicles in a dormant state. Interestingly, we saw that the addition of rapamycin resulted in fewer pAKT-labeled primordial and primary follicles. Other studies showed an increase in AKT activation following mTOR inhibition, possibly due to a feedback loop [40,57]. However, another study concluded that low concentrations of rapamycin are responsible for increased AKT activation via mTORC1 signaling, while higher concentrations of rapamycin resulted in decreased AKT phosphorylation mainly via the mTORC2 pathway, which could explain the results that we observed [58]. The mTOR pathway is additionally known for its function in cell proliferation [59]. Indeed, we observed that rapamycin was able to counteract cryopreservation-induced follicle proliferation. Furthermore, our results showed that even though the contact time between rapamycin and the ovary is relatively short during the freezing process, its positive effects can still be observed three weeks after transplantation.

To our knowledge, the addition of rapamycin during the cryopreservation process followed by transplantation has not yet been analyzed. Recent studies testing the inhibitor focused on the injection of rapamycin after transplantation [37,38,60]. However, a new vitrification protocol including pre-treating ovaries with rapamycin was tested. This study showed that rapamycin was able to inhibit mTOR pathway activation in ovaries directly after the thawing process and in ovaries five days after grafting in mice [61]. In addition, Chen et al. showed that the injection of rapamycin in mice resulted in lower levels of follicle proliferation [62].

Next, we found no apoptosis, DNA damage, or fibrosis in fresh ovaries and slow-frozen ovaries with or without rapamycin three weeks after transplantation. An explanation for this could be that three weeks is too long for analyzing these factors after transplantation. Indeed, double-stranded DNA breaks can be repaired in a few hours [63]. To more confidently state that there are indeed no effects on apoptosis, DNA damage, and fibrosis, these factors should be analyzed at a shorter time point after ovarian transplantation. However, our results indicate that rapamycin does not cause long-term tissue or DNA damage. Furthermore, CD31 analysis revealed no differences in blood vessel formation after transplantation between the three experimental groups.

Follicle quantification showed that the slow-freezing of ovaries followed by transplantation caused a lower primordial follicle pool. Indeed, a lower percentage and total amount of primordial follicles per mm^2^ were observed in slow-frozen ovaries compared to fresh ovaries. Ovaries cryopreserved with rapamycin showed no difference in terms of the percentage or number of primordial follicles per mm^2^ compared to cryopreservation without rapamycin. Cryopreservation with rapamycin resulted in a significantly lower percentage and number of primary follicles per mm^2^ compared to fresh ovaries, and a similar trend could be observed for the SF ovaries in the control medium. Furthermore, the percentage of secondary or more growing follicles was significantly higher when ovaries were slow-frozen before transplantation either with or without rapamycin in the freezing medium. One of the explanations for this could be that cryopreservation causes more primary follicles to develop into secondary or more growing follicles.

While these results showed that the addition of rapamycin to the freezing medium was not able to counteract a cryopreservation-induced decrease in primordial follicles, the quality of follicles in SF ovaries with rapamycin may be better compared to follicles in SF ovaries with the control medium. To investigate this, follicle health and quality should be examined and compared between SF ovaries with or without rapamycin.

In this study, we promisingly found that the addition of rapamycin to the freezing medium resulted in the lower activation and proliferation of primordial and primary follicles. As other research found promising results when injecting mice post-transplantation with follicle activation inhibitors, e.g., a recent study with AMH, the natural inhibitor, a follow-up study could try to combine the cryopreservation of ovaries, including rapamycin, with the injection of follicle activation inhibitors (e.g., AMH) and/or antiapoptotic/angiogenic agents to increase graft quality even more [34].

For our experiments, mice models were used. While the morphology of human and mouse reproductive systems are distinctive (e.g., the mouse has a bicornuate uterus compared to the human single uterus), their reproductive cycles are very similar [64]. Indeed, similarly to humans, the mouse reproductive cycle oscillates periodically via fluctuations in progesterone and estrogen concentrations, which shows that mice models are suitable for answering certain reproductive research questions [64]. In order to increase translatability, a xenograft study could be performed, transplanting human ovarian tissue into immune-deficient mice. Furthermore, as rapamycin is already used in the clinic, for instance, to prevent rejection after organ transplantation, starting the use of rapamycin during ovarian cryopreservation in humans should not be too difficult [35,36].

Our next aim was to compare the recently described ovarian transplantation model between the skin layers of the ears with the more conventional and invasive model under the kidney capsule. As existing ovarian transplantation models have different inconveniences, e.g., the need for invasive surgery or long periods of ischemia after transplantation, a less invasive model was developed, and in addition, ovaries transplanted between the skin layers of the ears remained easily accessible for local treatment injection and observational studies [12,13,14,55]. We previously showed that SF/thawed mice ovaries transplanted for either three days or three weeks into the ears of SCID mice were revascularized three weeks after transplantation, with an increased proliferation of cells. Furthermore, we found no modulation in apoptosis and a decrease in fibrosis three weeks after transplantation compared to three days [55]. These results indicated that this new ear transplantation model could be suitable for analyzing follicle activation and proliferation and for testing pharmacological strategies. In order to further complete this study, this new transplantation site was compared with transplantation under the kidney capsule by autotransplanting mouse ovaries to either site, followed by the injection of either LY294002 or vehicle control, or neither, and recovered three weeks later for IHC analysis.

Ovaries transplanted to either side without follow-up injections showed no differences in the primordial follicle pool. Furthermore, no apoptosis was observed for both transplantation sides, indicating no long-term effects of the transplantation progress on cellular death. CD31 staining demonstrated similar levels of vascular endothelial cells between the sites, showing that vascularization is comparable in ovaries transplanted to either site. Additionally, no differences in follicle proliferation and the activation of the mTOR pathway were found in ovaries transplanted for three weeks between the skin layers of the ears or under the kidney capsule. However, the activation of the AKT pathway was significantly lower in ovaries that were transplanted using the ear model compared to the kidney capsule model, indicating that primordial and primary follicles might be in a more dormant state after ovarian transplantation between the skin layers of the ear compared to transplantation under the kidney capsule. Taking these results together, it is established that the new non-invasive ear model is as suitable for ovarian transplantation as the more conventional kidney capsule model.

As one of the main advantages of the new ear model is that it provides easy accessibility for local treatment injections, we wanted to compare the effects of local treatment for ovaries transplanted between the skin layers of the ears with the effects of IP treatment for ovaries transplanted under the kidney capsule, followed by the recovery of the ovaries three weeks later. For the treatment, the strong PI3K inhibitor LY294002 was chosen [65]. In a previous in vitro organotypic ovarian culture study, our team showed that the culture-induced activation of the PI3K/PTEN/AKT pathway was reversed by the addition of LY294002 to the culture medium [40]. This led us to test this inhibitor in the comparison between the ear and kidney capsule transplantation model.

Body weight and mice behavior were monitored for three weeks after transplantation and injection in order to analyze the toxic side effects of LY294002. No difference in weight gain was observed when mice were injected with or without LY294002, either locally or IP, indicating that the inhibitor has no negative effects on the growth of mice. This is in agreement with another study, in which they injected xenografted BALB/C mice with LY294002 (IP) and found no effects on weight gain [66]. Behavioral analysis showed mice injected IP with LY294002 were significantly less active compared to the local injection of LY294002 in the ears and compared to mice injected with the control. This indicates that mice potentially suffer more from the IP injection of LY294002 compared to local injection. This could be explained by the fact that after local injection, the treatment mainly remains at the injection site and surrounding tissue, while IP injection causes the treatment to be systemically distributed, meaning that it could affect the entire body [67]. Furthermore, no differences in weight gain and behavioral score were observed between the two types of surgery. This suggests that even though the kidney model is more invasive, it does not affect the growth or behavior of mice differently compared to the less invasive ear model.

In order to examine whether the local and IP injections of LY294002 have similar or different effects on the follicle pool in transplanted ovaries, follicle density analysis was performed. This analysis revealed that the injection of LY294002 locally for the ear model and IP injection for the kidney capsule model had the same effect on follicle density both with respect to the percentage of primordial, primary, and secondary or more growing follicles compared to the total amount of follicles per section and the total number of each follicle type per mm^2^. Furthermore, no apoptosis and no DNA damage were observed in transplanted ovaries after injections with or without LY294002, for both the ear and kidney capsule models. Therefore, LY294002 showed no toxic long-term side effects on ovaries three weeks after transplantation and injection in both models. The injection of LY294002, both locally or IP, did not affect follicle proliferation and the activation of the mTOR pathway compared to the control injection in ovaries transplanted for three weeks. An explanation for this could be that the effects of LY294002 on proliferation and mTOR pathway activation are only possible in the short term, and three weeks after transplantation and injection is too long for detecting the effects. However, AKT activation was lower in mice treated with LY294002 via IP injections compared to the IP control injection. This effect was not observed for local injections. There is a trend that AKT activation is lower in ovaries with the local control injection compared to the IP control injection. This could explain why LY294002 seems to only have an effect for IP injection, as transplantation under the kidney capsule tends to activate the AKT pathway to a greater extent compared to transplantation between the skin layers of the ears.

## 5. Conclusions

Our results indicate that the addition of the mTOR inhibitor rapamycin during the OTCTP procedure was able to transiently maintain primordial follicles in a quiescent state. Limiting the massive follicle activation of the primordial follicle pool is a promising method that can improve the longevity of the ovarian graft for fertility restoration, which could prolong the time frame in which recovered patients could become pregnant, and thus increase the overall possibility for pregnancies after the use of OTCTP.

Furthermore, follicle activation and proliferation could be analyzed similarly when ovaries were transplanted between the skin layers of the ears and when transplanted under the kidney capsule. The injection of the PI3K inhibitor LY294002 locally for ovaries transplanted between the skin layers of the ears or IP for ovaries transplanted under the kidney capsule seemed to have similar effects on follicle proliferation and activation, with a decrease in side effects when injected locally, indicating that both sites could be used to test pharmacological strategies.

## Figures and Tables

**Figure 1 medicina-59-01474-f001:**
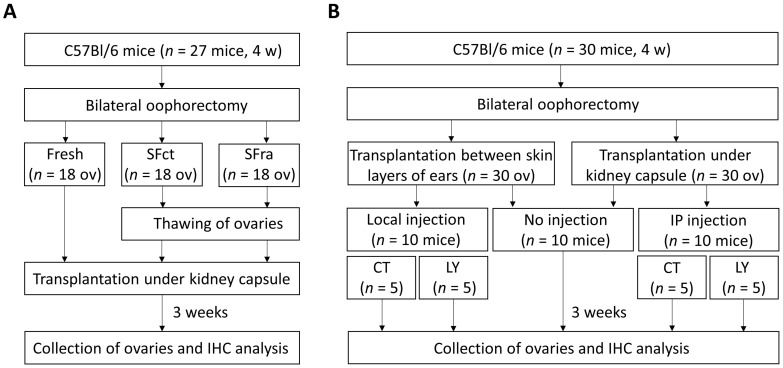
Experimental design for the mice model used to investigate the effects of adding rapamycin to the freezing medium on follicle activation and proliferation (**A**) as well as the comparison between two ovarian transplantation sites, namely, between the skin layers of the ears and under the kidney capsule (**B**). 4 w = 4 weeks old, Fresh = ovaries transplanted directly after oophorectomy, SFct = slow-freezing in control medium, SFra = slow-freezing with rapamycin, IHC = immunohistochemistry, IP = intraperitoneal, CT = control injection, LY = injection with LY294002.

**Figure 2 medicina-59-01474-f002:**
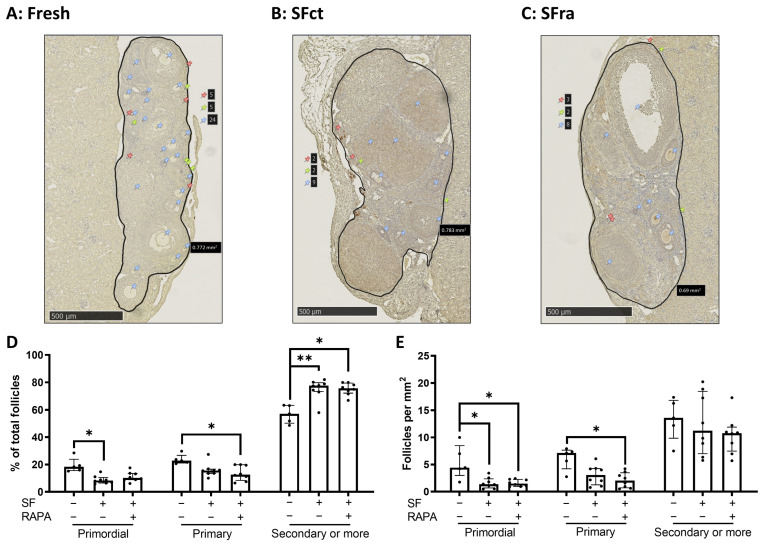
Follicle density assessment in fresh, SFct, or SFra mice ovaries autotransplanted under the kidney capsule of C57BL/6 mice (4 weeks old) for three weeks. Representative images of LHX8 staining of fresh (**A**), SFct (**B**), or SFra (**C**) mice ovaries transplanted under the kidney capsule. Red pins indicate primordial follicles, yellow pins primary follicles, and blue pins secondary or more growing follicles. Follicle density was either expressed in the percentage of primordial, primary, and secondary or more growing follicles relevant to the total amount per section (**D**) or as the total number of each follicle type per mm^2^ (**E**) (median with interquartile range). *n* = 5–8 ovaries per group. * *p* ≤ 0.05, ** *p* ≤ 0.01. RAPA = rapamycin.

**Figure 3 medicina-59-01474-f003:**
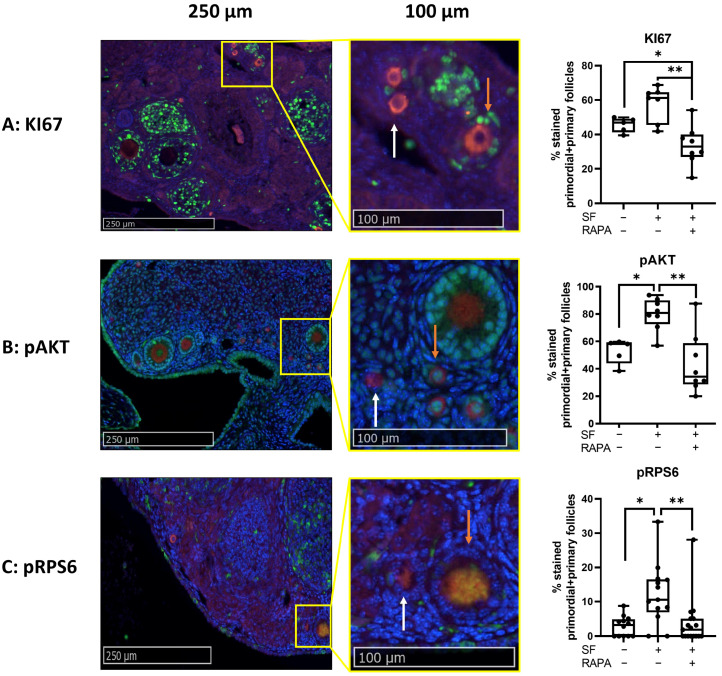
Effects of adding rapamycin to the freezing medium on follicle activation and proliferation in fresh, SFct, or SFra mice ovaries autotransplanted under the kidney capsule of C57BL/6 mice (4 weeks old) for three weeks. Immunohistochemistry (IHC)-assisted quantification (median + min to max) of the percentage of primordial and primary follicles labeled for KI67 (**A**), pAKT (**B**), and pRPS6 (**C**), including representative images of ovaries in the fresh group. Red staining = DDX4, green staining = KI67, pAKT, or pRPS6. White arrow = non-green-stained follicle, orange arrow = green-stained follicle. * *p* ≤ 0.05, ** *p* ≤ 0.01. SF = slow-frozen/thawed ovaries, RAPA = rapamycin. (**A**,**B**) *n* = 5–8 ovaries per group, (**C**) *n* = 12–17 ovaries per group.

**Figure 4 medicina-59-01474-f004:**
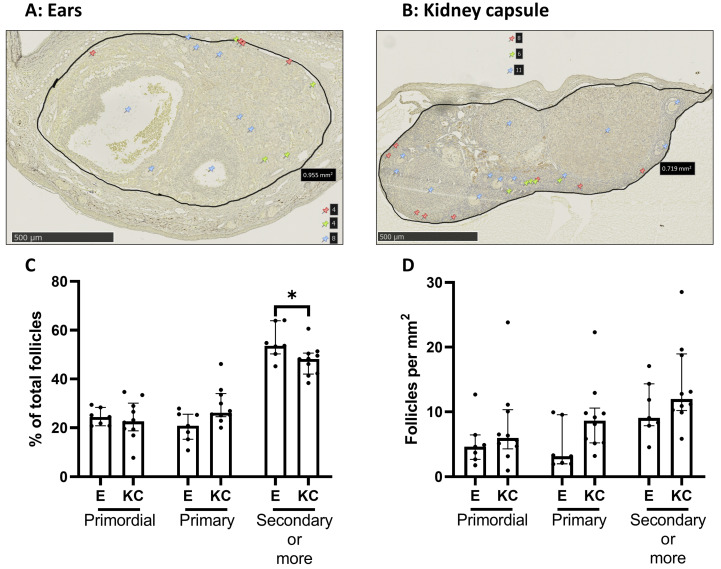
Comparison of follicle densities between fresh mice ovaries autotransplanted either between the skin layers of the ear or under the kidney capsule of C57BL/6 mice (4 weeks old). Representative images of LHX8 staining of fresh mice ovaries transplanted either between the skin layers of the ears (**A**) or under the kidney capsule (**B**). Red pins indicate primordial follicles, yellow pins primary follicles, and blue pins secondary or more growing follicles. Follicle density was either expressed in the percentage of primordial, primary, and secondary or more growing follicles relevant to the total amount per section (**C**) or as the total number of each follicle type per mm^2^ (**D**) (median with interquartile range). *n* = 7–10 ovaries per group. * *p* ≤ 0.05. E = transplantation of ovaries between skin layers of ears, KC = transplantation of ovaries under the kidney capsule.

**Figure 5 medicina-59-01474-f005:**
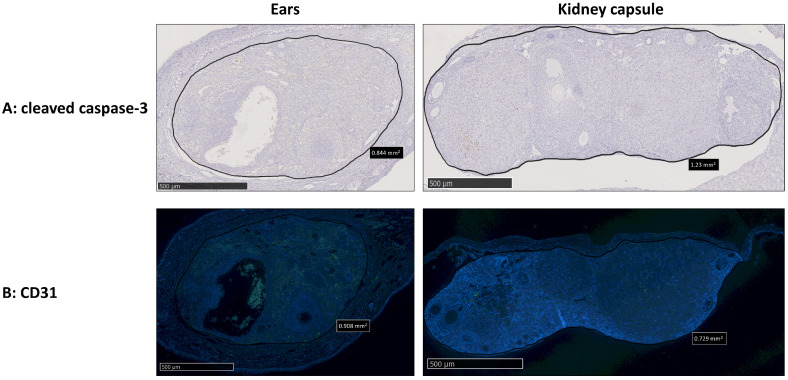
Comparison of apoptosis and vascular endothelial cells between fresh mice ovaries autotransplanted either between the skin layers of the ear or under the kidney capsule of C57BL/6 mice (4 weeks old). Representative images of cleaved caspase-3 (**A**) and CD31 (**B**) staining of fresh mice ovaries transplanted either between the skin layers of the ears or under the kidney capsule. Green staining = CD31. *n* = 7–10 ovaries per group.

**Figure 6 medicina-59-01474-f006:**
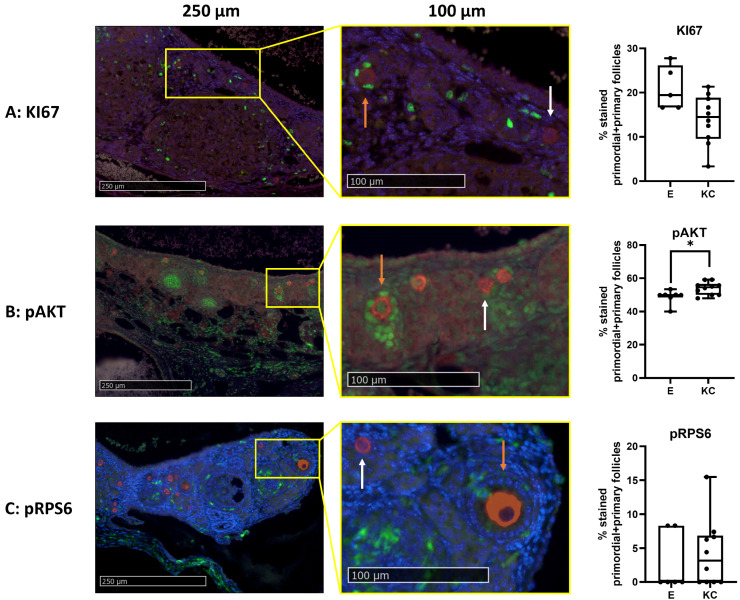
Comparison of follicle activation and proliferation between fresh mice ovaries autotransplanted either between the skin layers of the ear or under the kidney capsule of C57BL/6 mice (4 weeks old). IHC-assisted quantification (median + min to max) of the percentage of primordial and primary follicles labeled for KI67 (**A**), pAKT (**B**), and pRPS6 (**C**), including representative images of ovaries transplanted under the kidney capsule. Red staining = DDX4, green staining = KI67, pAKT or pRPS6. White arrow = non-green-stained follicle, orange arrow = green-stained follicle. * *p* ≤ 0.05. *n* = 5–10 ovaries per group. E = transplantation of ovaries between skin layers of ears, KC = transplantation of ovaries under the kidney capsule.

**Figure 7 medicina-59-01474-f007:**
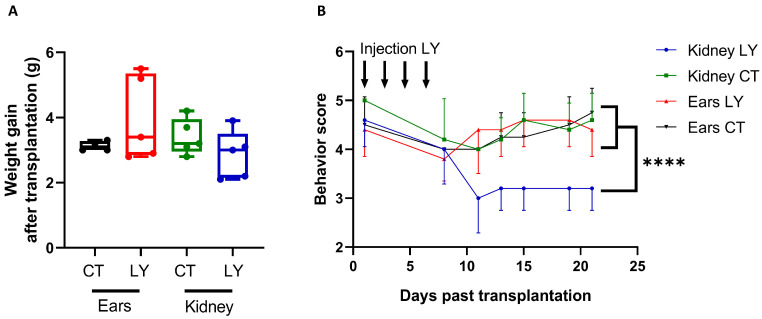
Effect of LY294002 (LY) injection on weight gain and mice behavior after ovarian autotransplantation into C57BL/6 mice (4 weeks old). Ears were injected locally and the kidney capsule was injected intraperitoneally. Weight gain (in grams; median + min to max) (**A**) and behavior score (0 = least active, 5 = most active; mean ± SEM) (**B**) of treated or control mice monitored for three weeks after transplantation. *n* = 4–5 mice per group. **** *p* ≤ 0.0001. CT = vehicle control, LY = LY294002.

**Figure 8 medicina-59-01474-f008:**
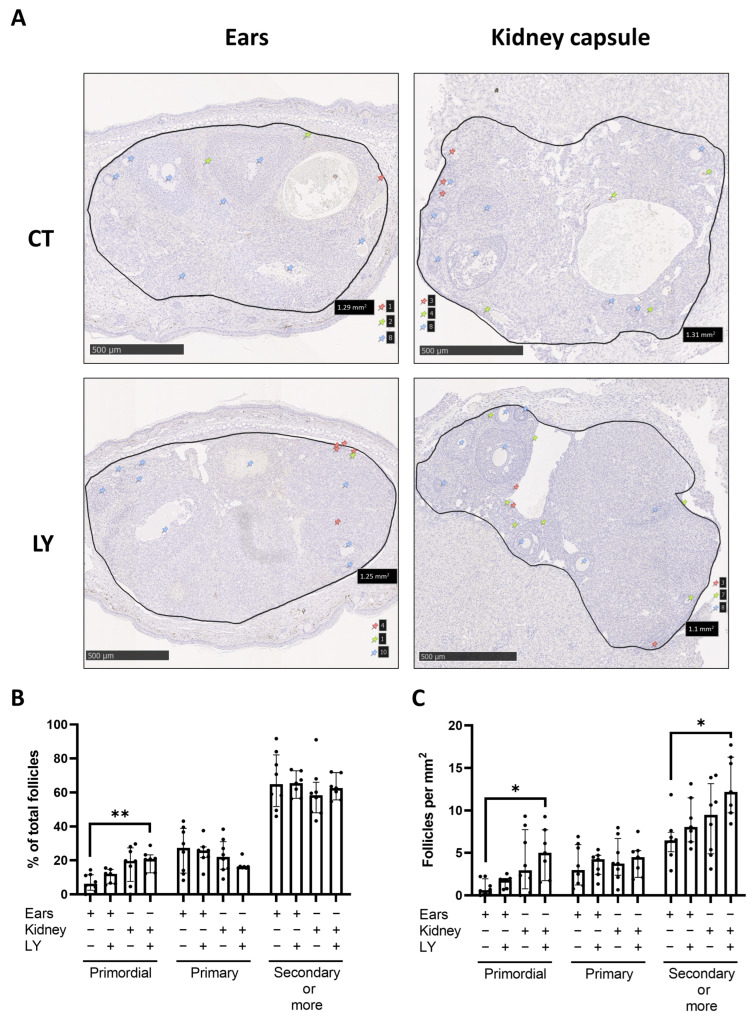
Comparison of follicle densities for LY injection after ovarian autotransplantation into C57BL/6 mice (4 weeks old) either locally in the ears or intraperitoneally (IP) when ovaries were transplanted under the kidney capsule. (**A**) Representative images of LHX8 staining of fresh mice ovaries transplanted either between the skin layers of the ears or under the kidney capsule, followed by local injection with LY or vehicle control for ovaries transplanted between skin layers of the ears, or IP for ovaries transplanted under the kidney capsule. Red pins indicate primordial follicles, yellow pins primary follicles, and blue pins secondary or more growing follicles. Follicle density was either expressed in the percentage of primordial, primary, and secondary or more growing follicles relevant to the total amount per section (**B**) or as the total number of each follicle type per mm^2^ (**C**) (median with interquartile range). *n* = 7–8 ovaries per group. * *p* ≤ 0.05, ** *p* ≤ 0.01. CT = control injection, LY = injection with LY.

**Figure 9 medicina-59-01474-f009:**
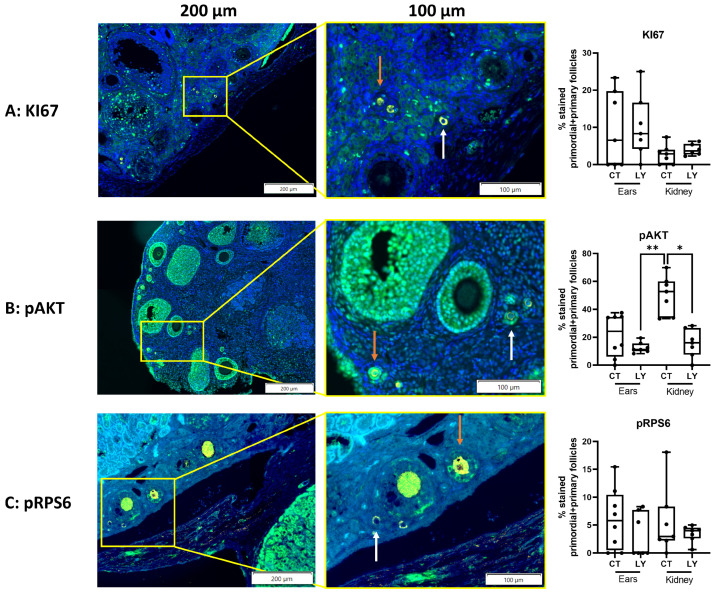
Comparison of follicle activation and proliferation for LY injection after ovarian autotransplantation into C57BL/6 mice (4 weeks old). Ears were injected locally and the kidney capsule was injected intraperitoneally. IHC-assisted quantification (median + min to max) of the percentage of primordial and primary follicles labeled for KI67 (**A**), pAKT (**B**), and pRPS6 (**C**), including representative images of ovaries transplanted under the kidney capsule followed by control IP injection. Yellow staining = DDX4, green staining = KI67, pAKT, or pRPS6. White arrow = non-green-stained follicle, orange arrow = green-stained follicle. * *p* ≤ 0.05, ** *p* ≤ 0.01. *n* = 6–8 ovaries per group. CT = vehicle control, LY = LY294002.

**Table 1 medicina-59-01474-t001:** Mice behavioral score conditions. Adapted from Herrmann et al. [51].

Behavioral Score	Score Condition
0	No activity; breathing issues; death anticipated
1	No activity; no food intake
2	Low active state and sleepy; unreactive to human interaction
3	Low to normal active state; reactive to human interaction
4	Normal active state and fast; tries to escape when scruffed
5	Very active state, strong and fast; agitated when scruffed

## Data Availability

The corresponding author will provide the data underlying this article upon reasonable request.

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
