# Peer review of "The mTOR Inhibitor Rapamycin Counteracts Follicle Activation Induced by Ovarian Cryopreservation in Murine Transplantation Models"

_medicina, 2023, doi:10.3390/medicina59081474_

Round 1

Reviewer 1 Report

1. Use of Western blot for confirmation: It would be valuable to address why Western blot analysis was not employed in this study to confirm the findings. Since the authors have utilized Western blot in a previous study, it is important to provide a rationale for the decision to exclude it in this particular investigation. This clarification would enhance the scientific rigor and ensure consistency in the methodology.

2. Improvement of graph quality: The manuscript should aim to improve the quality of the graphs presented. Clear and visually appealing graphs enhance the readability and interpretation of the data. Authors should consider adjusting the resolution, font size, and clarity of the figures to ensure that the information is easily understandable.

3. Consider using a graphical abstract: Instead of flowchart tables, the authors may consider utilizing a graphical abstract to visually depict the experimental design. A graphical abstract can provide a concise overview of the study, including key methods and findings, which can help readers quickly grasp the main aspects of the research.

4. Importance of the findings: The manuscript briefly mentions the promising way of improving the longevity of the ovarian graft through the addition of rapamycin during the OTCTP procedure. It would be beneficial to elaborate on the potential clinical implications and significance of these findings. How might this contribute to the field of fertility preservation and prepubertal patients' reproductive health?

5. Quantifying method and software: The manuscript should provide detailed information about the quantifying method used for the figures and the specific software employed for data analysis. It is important to describe the image analysis techniques used to measure and quantify the experimental results accurately. Additionally, specifying the software used for data analysis and any specific algorithms or parameters applied would enhance the transparency and reproducibility of the study.

Minor editing of English language required

Reviewer 2 Report

The manuscript entitled "The mTOR inhibitor rapamycin counteracts follicle activation 2 induced by ovarian cryopreservation in murine transplantation 3 models" is well written and covers an important area which belongs to the fertility preservation of prepubertal oncopatients. However in general manuscript is well I have some questions and recommendations.

1. In the introduction it should be more clearly indicated how precisely cryopreservation initiates ovatian follicle premature activation.

2. Provide in the Introduction section some scheme illustrating interaction of mTOR, Akt, PTEN, PI3k and other components involved in follicular activation. Also indicate at this scheme points of pathways which you affect using inhibitors.

3. In the current article only types of heterotopic transplantation are investigated, however in case of human traditionally orthotopic type of transplantation is used. Please, add another group of animals with orthotopic transplantation.

4. In the pictures given in the manuscript evidently less number of follicles is observed in eyes rather than in the kidney. Discuss it. Probably, the kidney capsule is the better site for transplantation rather than the ear skin?

5. Most simple way to avoid premature ovarian follicles activation to have the possibility to take children is to no perform transplantation of all cryopreserved tissue simultaneously. For example, first pack of tissue can be transplanted before first pregnancy and the second pack before the second pregnancy. And no one treatment of some specific agents is required. Discuss it.

6. Describe in details protocol of ovaries fixation in hystology section.

7. Check in the text "in vivo" and "in vitro". It should be italic, correct.

8. It is not clearly indicated how the number of follicles per ovary (part of ovary?) has been counted? Some ovaries are big and if the section is 5 micrometers these ovaries could be counted twice if the number of follicles were counted across all the slides of the transplanted piece. 

9. How big were pieces of ovaries used for cryopreservation and transplantation? lenght x depth x width

10. Also you may discuss another potential approaches for fertility preservation in prepubertal girls, for example, ex utero ovarian follicles in vitro culture (Filatov et al., 2015. Acta Naturae) and their limitations.
